# Early Alzheimer’s Disease Detection: A Review of Machine Learning Techniques for Forecasting Transition from Mild Cognitive Impairment

**DOI:** 10.3390/diagnostics14161759

**Published:** 2024-08-13

**Authors:** Soraisam Gobinkumar Singh, Dulumani Das, Utpal Barman, Manob Jyoti Saikia

**Affiliations:** 1Faculty of Computer Technology, Assam down town University, Guwahati 781026, Assam, India; gobinkumarsoraisam@gmail.com (S.G.S.); dean.foct@adtu.in (U.B.); 2Biomedical Sensors and Systems Lab, University of North Florida, Jacksonville, FL 32224, USA; 3Department of Electrical Engineering, University of North Florida, Jacksonville, FL 32224, USA

**Keywords:** Alzheimer’s disease, mild cognitive impairment, AD, MCI, MRI, PET, machine learning

## Abstract

Alzheimer’s disease is a weakening neurodegenerative condition with profound cognitive implications, making early and accurate detection crucial for effective treatment. In recent years, machine learning, particularly deep learning, has shown significant promise in detecting mild cognitive impairment to Alzheimer’s disease conversion. This review synthesizes research on machine learning approaches for predicting conversion from mild cognitive impairment to Alzheimer’s disease dementia using magnetic resonance imaging, positron emission tomography, and other biomarkers. Various techniques used in literature such as machine learning, deep learning, and transfer learning were examined in this study. Additionally, data modalities and feature extraction methods analyzed by different researchers are discussed. This review provides a comprehensive overview of the current state of research in Alzheimer’s disease detection and highlights future research directions.

## 1. Introduction

Alzheimer’s Disease (AD) is a specific type of dementia characterized by memory loss, cognitive decline, and behavioral changes. Over time, these symptoms become severe enough to disrupt the cognitive functions of a person’s mental capabilities [1]. The World Health Organization (WHO) [2] estimates that the global population aged 60 and older reached 1 billion in 2020. This number is projected to double to 2.1 billion by 2050. According to the Longitudinal Aging Study in India (LASI) conducted from 2018 to 2020, dementia affects an estimated 7.4% of Indian adults aged 60 and above, amounting to over 8.8 million individuals from a population of 1.37 billion people [3]. Initially, patients develop Mild Cognitive Impairment (MCI)and then potentially progress to AD, but not all MCI patients go on to acquire AD [4]. MCI is a transitional stage between normal aging and dementia. Individuals with MCI experience memory loss and other cognitive difficulties more severe than expected for their age, but not severe enough to interfere with daily activities. MCI is not a definitive diagnosis of AD, but it increases the risk of developing dementia. Treatment strategies and medications are generally more effective early in the development of AD [5], making it necessary to take early action to reduce the progress of the disease since there is currently no known effective cure for AD [6]. Therefore, the primary emphasis of the ongoing research is on predicting the progression from MCI to AD.

There are many biomarkers that are used to detect AD such as neuroimaging, biological, and genetic which include fMRI [7] which is commonly used for studying brain functions and activities by measuring changes in blood flow, while other techniques like PET, EEG, MEG, fNIRS, and TMS are also used for similar purposes for the brain–computer interfaces [8,9,10]. The utilization of MRI images is highly prevalent in the early identification and categorization of AD.

The accurate identification of AD-affected areas through MRI is a key focus of this research, as MRI serves as a valuable diagnostic tool for detecting plaques and affected regions associated with AD [11,12]. Nowadays, the adoption of multi-modality studies involves employing more than one modality for each subject. This approach enables the extraction of features from different modalities, potentially encompassing complementary information. For instance, ADAS-Cog, MRI, MMSE, FDG-PET, and CSF are often used modalities.

In some publications, publicly accessible datasets have become increasingly valuable in the research field, offering a variety of biomarker information, including neuroimaging modalities, genetic and blood data, as well as clinical and cognitive assessments [13,14]. Some of the most widely utilized datasets include the Alzheimer’s Disease Neuroimaging Initiative (ADNI) [15] (https://adni.loni.usc.edu/ (accessed on 25 October 2023)adni.loni.usc.edu), the Australian Imaging, Biomarker and Lifestyle Flagship Study (AIBL) (www.aibl.csiro.au (accessed on 25 October 2023)) [16], the Open Access Series of Imaging Studies (OASIS) (www.oasis-brains.org (accessed on 27 October 2023)) [17], and Minimal Interval Resonance Imaging in Alzheimer’s Disease (MIRIAD) [18]. Additionally, the publicly available database for clinical Alzheimer’s data from longitudinal studies in Japan, the Japanese Alzheimer’s Disease Neuroimaging Initiative (JADNI) database, provides valuable resources for research [19,20].

It has been acknowledged that machine learning techniques are widely employed for AD diagnosis [21,22]. In the context of medical Science, ML models have emerged as valuable tools for classifying and predicting abnormalities [23,24]. Researchers primarily directed their attention toward longitudinal data collected over multiple follow-up visits to identify patterns and parameters that evolve over time and are indicative of progression from MCI to AD [25]. Detecting AD patients in advance allows precise diagnosis using MRI and other medication strategies that can potentially slow down the disease’s progression, providing better health outcomes for patients and their future lives [23,24,25].

Early identification of the person at risk for AD during the MCI stage allows for timely intervention and treatment, but there are certain things that need to be considered in the MCI stage. Distinguishing between progressive Mild Cognitive Impairment (pMCI) patients and sMCI patients [23,24,25]. pMCI refers to individuals with MCI who are likely to progress to AD within a relatively short period, usually within a few years. These individuals exhibit a decline in cognitive function over time. Stable Mild Cognitive Impairment (sMCI) refers to individuals with MCI whose cognitive function remains relatively stable over time. They may experience mild cognitive difficulties, but these do not worsen significantly. There is an overlap in certain characteristics between pMCI and sMCI patients, such as brain region shrinkage [23]. Identifying the biomarkers and features that can distinguish between the two is a challenging task [23,24,25]. Also generalizing characteristics from various data modalities is difficult [24,25]. To address these challenges, ML algorithms have been employed to identify hidden patterns and relevant features that differentiate between pMCI and sMCI patients [25,26,27]. These algorithms serve as valuable aids for physicians in making critical predictions about sMCI and pMCI [25,26,27].

In recent years, the prevalent classification techniques in Machine Learning (ML)and Deep Learning (DL) include Logistic Regression (LR), Support Vector Machine (SVM), Random Forest (RF), Ensemble Learning (EL), Artificial Neural Network (ANN), DL, and Transfer Learning (TL). The quality of an SVM is determined by the optimization of a problem that has a globally optimum solution [28]. In the process of feature extraction, the significance of both SVM and ANN becomes evident, which underscores the utility of combining neural networks and intelligent agents for the analysis of medical images [29]. Certain researchers have also employed ensemble methods, which involve utilizing a collection of decision-making systems that implement diverse strategies to combine classifiers that enhance the predictions on new data and improve the overall accuracy of classification and prediction for AD [30,31]. However, DL is useful when the dataset is very large and unstructured data. It is used in medical images to automatically identify and select relevant features or patterns and then, to use as input for neural networks [32,33]. TL has the advantage that the knowledge gained from one domain improves the performance of a neural network in a related domain, ultimately saving time and resources [34,35].

This review provides the forecasting of the early transition from MCI to AD through the application of ML and DL models. It summarizes the findings and focuses on studies that employed longitudinal data for MCI to AD prediction. Not only these, it also shows various aspects of those studies, including participant demographics, features selected, longitudinal follow-up data, ML and DL classifiers, performance metrics, and critique. The objective is to offer a thorough summary of the landscape of MCI to AD conversion research, aiding new researchers in understanding the field and identifying their research directions.

## 2. Review Procedure and Research Approach

In this part, a detailed account of the review procedure utilized in this article is presented. The process initiates with the formulation of a set of research questions. These inquiries serve as the foundation for the systematic exploration of literature pertaining to the detection of AD through ML and DL techniques.

### 2.1. Inquiry Objectives

The primary objective of this study is to offer a comprehensive summary of the present advancements in the prompt identification of individuals transitioning from MCI to AD through the application of ML and DL methodologies. To fulfill this goal and provide a thorough examination of diverse techniques employed in AD detection, the search is guided by specific objectives. These objectives aim to investigate critical aspects, including the geographical distribution of involvement in the study, the data modalities utilized, methodologies for taking relevant characteristics out of unstructured neuroimaging data, and the classifiers implemented in forecasting individuals with sMCI and pMCI.

Certainly, here are the research questions:What are the geographical origins (country) of the dataset of the individuals involved in the research studies?Which types of data modalities are employed as features?How many of the studies incorporate data obtained through longitudinal follow-up, and what is the time frame of the data follow-up considered in the literature?What spectrum of transitions from MCI to AD is recognized and documented in the reviewed papers?What is the number of studies that explored the utilization of ML techniques to predict the early transition from MCI to AD?How many DL and TL models have been used in the literature to predict the early conversion of MCI to AD?Under what specific conditions or circumstances do the ML and DL models tend to perform better in predicting MCI to AD conversions?

### 2.2. Search Strategy

A thorough search was carried out using Google Scholar (https://scholar.google.co.in/ (accessed on 24 September 2023)), IEEE Xplore, Scopus, WOS, PubMed, MDPI, and ScienceDirect to find noteworthy articles on the subject. In addition to these choices, the review paper adheres to PRISMA’s requirements [36]. The research studies taken into consideration for this review have used specialized feature selection strategies, multi-task learning, transfer learning, multi-kernel learning, SVM, and ANN in combinations. This study has also included relevant publications on deep learning for Alzheimer’s. Notably, papers utilizing traditional ML such as SVM, Naive Bayes (NB), Binary Logistic Regression (BLR), Gaussian Mixture Model (GMM), k-Nearest Neighbors (kNN), k-means were from the years 2015 to 2023, Multi-Layer Perceptron (MLP), Convolutional Neural Network (CNN), Deep Neural Network (DNN), and TL papers were from 2018 to 2023. The search strategy used in this literature review is depicted in Figure 1.

After a thorough study, inclusion and exclusion criteria were applied at various stages. Initially, duplicate papers were eliminated from the results. Subsequently, papers were screened based on the information provided in their abstracts. The focus of the study was mainly on those papers that utilized longitudinal data of MCI to AD, as these studies analyze follow-up data to make conclusive decisions regarding the classification of pMCI and sMCI patients. Consequently, investigations relying solely on cross-sectional data to categorize patients with pMCI and sMCI were omitted. For inclusion in the review, papers were required to provide a clear explanation of performance metrics. Studies with explicit descriptions of performance metrics, including but not limited to accuracy, ROC, AUC, sensitivity, and specificity, were selected for inclusion in the review. This rigorous selection process ensures that the chosen studies meet the criteria necessary for a comprehensive and reliable review.

## 3. Results

A thorough presentation of the outcome data, providing a comprehensive summary of the results acquired from the performed study, is provided in this part. Based on the literature study, the discussion can be categorized into two sections: the first discussion is based on traditional ML and the second discussion is based on DL and TL for classifying MCI to AD. An explanation of the conventional ML models used in literature for classifying MCI to AD is provided in Table 1. ML algorithms play a crucial role in understanding and predicting cognitive decline. In the first category of discussion, SVM with linear and radial basis kernels, are the most commonly used classification models to identify hyperplanes and non-linear patterns, respectively, contributing to binary classification tasks [37,38,39,40,41,42,43,44,45,46,47,48,49]. Non-parametric classification techniques, such as kNN, offer flexibility in capturing intricate relationships without making explicit data distribution assumptions [50]. NB leverages probabilistic considerations for modeling the probability of MCI or AD based on specific features [40]. Reducing overfitting and handling intricate interactions are two advantages of RF [40]. The Cox proportional hazards model and its application in the Cox proportional model contribute to survival analysis, examining the time of progression from MCI to AD [51,52]. Regularized LR incorporates regularization to enhance predictions [44], while Lasso aids in feature selection and regularization [53,54]. GMM, kNN, extreme learning machines, and isolation forests contribute to a comprehensive tool for understanding and addressing the MCI to AD progression [50]. Additionally, both multi-modality and single-modality approaches in analyzing ADNI data for MCI to AD progression using LR, highlighting differences in data complexity, and feature selection methods shown in [55,56].

From Table 1, it is found that conventional ML methods offer some advantages in their ease of interpretability when applied to well-structured data with explicit feature engineering. Researchers demonstrated efficiency in such scenarios with limited datasets. In most of the cases, the dataset considered was Alzheimer’s disease Neuroimaging Initiative (ADNI) dataset with MRI images. The follow-up data considered in these studies were of 3 years in the majority of cases, where 6 times follow-up was performed in an interval of 6 months from screening. However, these approaches come with their set of drawbacks, necessitating extensive feature engineering and preprocessing efforts.

K. Ritter et al. [37] clearly outlined the objectives emphasizing the use of longitudinal data and proposed a supervised, non-parametric method for the prediction of MCI and AD. However, more extensive discussion is required on the challenges and potential biases introduced by the limited dataset size. Analyzing the approach with a larger population will be beneficial rather than being limited to the dataset (ADNI) used. L. Frolich et al. [38] highlighted the comprehensive combination of biomarkers to predict AD from MCI. SVM was used with bootstrapping methods to enhance the classifier and ROC was used to evaluate and compare the performance of classifiers, with an imbalanced class distribution. However, it was necessary to have a discussion on the challenges and potential effects of a limited dataset size. The utilization of k-cross validation for smaller datasets was also deemed necessary in order to prevent the occurrence of overfitting. Undoubtedly, the adoption of alternative measuring metrics beyond the Receiver Operating Characteristic (ROC) curve could offer considerable advantages in terms of simplification for their work. X. Long et al. [39] conducted an extensive data analysis that involved multiple regions of interest (ROIs) and compared their classification performance, contributing to a comprehensive analysis. This enhanced understanding of the algorithm’s effectiveness across different brain regions offered valuable insights into the morphological changes associated with AD. However, this work also needed to address the challenges and potential effects of limited dataset size. It was necessary to display feature selection steps. Additionally, it would be helpful to address potential confounding factors or methodological constraints in more detail. The article by T. Pereira et al. [40] has utilized time windows to analyze neuropsychological data across consecutive visits and captures individual trajectories of cognitive decline rather than relying on static MRI scans, potentially leading to improved prediction and valuable insights into disease progression. The longitudinal data analysis helped to track individual changes over time, offering more information about disease progression than single time-point analyses. This data-driven methodology is useful for extracting complex patterns and relationships from the data. However, further efforts are needed to fully understand how the models make their predictions using the time window approach. The dataset used in the study is from a Portuguese hospital, but a more generalized dataset is required. Zhao et al. [41] proposed a novel approach to predicting MCI progression using individual metabolic networks derived from longitudinal FDG-PET scans. This method capitalizes on the dynamic nature of brain metabolism, which is a critical factor in MCI progression. This study identified specific network features that are essential for predicting progression, providing valuable insights into the underlying mechanisms. However, the study’s small sample size may limit the generalizability of the findings. To address this, k-fold cross-validation could be employed to prevent overfitting. Additionally, the method used to construct the networks could be refined to incorporate additional information, such as anatomical constraints or temporal dynamics. K. Liu et al. [51] proposed a novel Independent Component Analysis (ICA) model in combination with Cox. The ICA approach aims to identify brain networks as data-driven, avoiding the limitations of a priori region-of-interest (ROI) analyses. This comprehensive approach captures both structural and functional aspects of the brain. However, it is important to discuss the challenges and potential effects of the limited dataset size. To enhance the analysis, direct fusion of multi-modal data using advanced techniques such as Joint ICA could be beneficial. Z. Sun et al. [42] presented a comprehensive methodology involving non-rigid registration, anatomical development representation; support vector field (SVF) for parallel transport normalization, and classification using SVM. This study has also provided detailed experiments and group-wise analysis. Although various features were used for anatomical development, it would be beneficial to include a detailed explanation of their specific relevance in MCI to AD conversion. The complicated suggested approach relies on the ADNI dataset but requires generalizability to diverse populations. G. Gavidia-Bovadilla et al. [43] combined CSF biomarker analysis and longitudinal MRI data, using systematic data analysis such as statistical analyses, including bimodal distribution observations, cut-off value confirmations, and the identification of variant (vr) and quasi-variant (qvr) regions of interest (ROIs). This study would be more instrumental if it showed CSF observations at later stages, substantial missing MRI data, and the specific population being research participants. Additionally, it would be helpful if the paper provided an interpretation of the models and an understanding of the features contributing to predictions. M. Gomez-Sancho et al. [44] conducted a comprehensive evaluation of various feature representations, including volumetric, voxel-based, and region-based features, using two classifiers (SVM and RLR) and repeated k-fold validation to mitigate overfitting. This study also addressed the class imbalance issue and its potential impact on specificity values. *p*-values for hypothesis testing (pAge, pHippo, pClass) were provided to assess the significance of differences between feature sets and classifiers. Although SVM tends to overpopulate the pMCI class, the study showed how this behavior can be mitigated. K. Kauppi et al. [52] proposed an innovative approach to predicting MCI to AD by integrating volumetric MRI, cognitive tests, and the Polygenic Hazard Score (PHS). The study utilized linear mixed-effect models and survival analysis to enhance the research. The findings were particularly relevant in the context of assessing individual risk for AD progression. Moreover, potential sources of bias, such as recruitment from memory clinics, would have strengthened the transparency of the research. T. Shen et al. [45] aimed to estimate the likelihood of MCI converting to AD in a year using SVM for classification and CNN for feature extraction from MRI images. The inclusion of preprocessing steps, such as segmentation and spatial standardization, added novelty to the methodology. However, the lack of detailed explanations of the variables and their significance in CNN and SVM was a limitation. Detailing the hyper-parameters (e.g., learning rate, kernel types) and tuning in CNN would have enhanced the study. S.H. Hojjati et al. [46] presented a study that integrated the approaches of rs-fMRI and sMRI and conducted a comprehensive evaluation of the performance of single-modality and two-modality approaches in MRI. Although specific atlases were used for rs-fMRI and sMRI, the selection of these atlases lacked a thorough justification. Additionally, this study did not discuss the reproducibility of findings or the generalizability of the classification model and follow-up procedure. C.C. Luk et al. [55] introduced a 3-Dimensional, whole-brain texture analysis that examined voxel intensities that may not be apparent through visual inspection. This work suggested a multi-factorial predictive model that combines MRI texture features with other clinical factors such as APOE-e 4, and genotype. This approach may enhance the predictive accuracy compared to using individual factors alone. However, visual aids or clear explanations, especially those related to texture changes, might have been helpful for readers. It would have been beneficial to explicitly discuss other potential limitations of the study, such as any inherent biases in the dataset or challenges in the texture analysis method. W. Zheng et al. [53] proposed a Multi Feature-Based network (MFN) that integrated multiple morphological features and included network properties, which added depth to the analysis. This work also provided detailed structural information and comprehensive feature integration. However, replication on larger, independent samples would strengthen the validity of the MFN, especially when applied to real-world situations for diagnosis. The exclusion of hippocampal and subcortical regions, which are crucial in the development of AD, could hinder the MFN’s performance. J.E. Arco et al. [47], combines two data modalities, MRI and neuropsychological tests, using searchlight analysis, which surpasses PCA in both uni-modal and multi-modal classification. Searchlight provides valuable insights into specific brain regions associated with AD progression. The use of nested cross-validation ensures optimal results. However, the results may not be generalizable to larger populations due to the small sample size. Additionally, the study only focuses on converters and not on non-converters, which may overlook valuable information about different stages of MCI or AD. E.E. Bron et al. [48] explicitly stated its objective and employed both SVM classifiers and CNN approaches. The comparison between the PND multi-center dataset and the Alzheimer’s Disease Neuroimaging Initiative (ADNI) dataset is highlighted in this work. External validation results are mentioned, but there is a lack of specific metrics and detailed analysis to quantify and explain the drop in performance. P.M. Rossini et al. [49] adopted a multidisciplinary approach, integrating graph analysis tools and ML methods to explore distinctive features and especially focusing on EEG data, adding depth to the research. This work required providing the information of baseline and follow-up of the dataset. It required specifying the exact number of samples or subjects used for prediction and validation for the limited dataset. M. Inglese et al. [54] developed a novel MRI-based radiomic predictive Vector (ApV) for AD diagnosis, employing an unsupervised approach. This study has shown robustness, repeatability across MRI scans, and potential for clinical application and future integration into clinical decision support systems. The lack of extensive external validation on diverse datasets limits the generalizability of the findings. Including individuals with Front Temporal Dementia (FTD) and Parkinson’s Disease (PD) in the control group for training the ApV raises concerns about potential biases. The observed limitations in performance at higher magnetic field strengths (3T) pose a significant constraint on the method’s applicability. The computational effort is required for preprocessing structural MRI data, particularly the segmentation step. Apart from these, it required baseline and longitudinal data with follow-up. S. Liu et al. [50] used a novel approach to ND techniques for predicting MCI conversion. A thorough evaluation including an ANN CV-based approach and hyper parameter optimization enhances the results. It also compares ND techniques with supervised binary classification algorithms (SVM and RF), providing a benchmark for this work. This study required a generalized dataset for data source limitation. The decision to consider only total scores for each assessment neglects potentially valuable details within the assessments. All modalities contribute equally to the prediction, which may not be true so a sensitivity analysis or feature importance assessment would provide the relative importance of different modalities. S. Park et al. [56] used brain imaging data, transitioning from one distinctive feature of the suggested approach, which is the diagnostic to prognosis time. To extract features from MRI data into gray matter (GM), white matter (WM), and CSF, this work used a thorough approach. Additionally, it contains areas of the brain linked to increased AD coefficients. The article required the size and characteristics of the original dataset from ADNI. While this work mentioned the use of logistic regression for the classification, it lacks specific details about the algorithm’s parameters. This work mentioned the AUC for performance evaluation; it would be beneficial to include additional metrics such as specificity and sensitivity. This work also mentioned transitioning from diagnosis to prognosis, but it is not entirely clear if the dataset uses longitudinal. It would be beneficial to include multiple time points for each subject and details of the temporal aspects of the data.

From the study of conventional ML, it was observed that the most commonly used model was SVM, either as single-modality or multi-modality. There was a variation in accuracy with SVM due to different features and feature selection techniques considered. In Figure 2, an analysis of different accuracies obtained on ADNI MRI data using SVM by different researchers is depicted.

It is observed from the study of traditional ML models to classify AD from MCI that these models have some limitations. These methods heavily rely on manually crafted features, which can be time-consuming and require domain expertise. Selecting the most informative features is crucial for model performance, but it can be challenging and subjective. Traditional ML algorithms might not capture complex patterns in the data, especially when dealing with high-dimensional and nonlinear relationships. This limitation can affect their performance compared to deep learning models in certain scenarios. These models can be sensitive to outliers in the data, which can significantly impact their performance. Robustness to outliers is essential for reliable AD diagnosis.

Table 2 provides an explanation of the application of deep learning and transfer learning algorithms for classifying MCI to AD, contributing to a thorough and comprehensive analysis. The different deep and transfer learning algorithms used in the literature are discussed in this section. Multi-Domain Transfer Learning (MDTL) enhances model generalization by transferring knowledge from multiple domains [57], while Deep Neural Network (DNN) employs a deep architecture to extract intricate features, providing an understanding of MCI to AD progression [58,59]. Convolutional Neural Network (CNN) specializes in image data, capturing spatial dependencies in neuroimaging and revealing structural changes relevant to disease conversion [60,61,62]. Deep Multi-Instance Learning is crucial for scenarios with partial patient information [63]. Group Factor Analysis identifies shared factors across patient groups, aiding in discovering common features indicative of MCI to AD conversion [64]. Recurrent Neural Network (RNN) effectively captures temporal dependencies in sequential data, valuable for understanding cognitive decline progression [59,61,65]. Graph Encoder and Variation Autoencoder integrate graph structures and unsupervised learning, capturing complex relationships within the data [65]. Long Short-Term Memory (LSTM) networks are designed for sequential data, modeling long-term dependencies in MCI to AD progression [66]. 3D ResNet efficiently captures spatial patterns in volumetric data like brain scans, and MoCo (Momentum Contrast) enhances feature representations through unsupervised learning, aiding in the discovery of subtle patterns indicative of disease conversion [67]. Unsupervised Learning allows the model to learn patterns without explicit labels, contributing to a comprehensive understanding of underlying structures. Zero-shot learning enables generalization to unseen classes, accommodating variations in MCI to AD manifestation while DsAN (Dual-Sparse Attention Network) incorporates sparse attention mechanisms to focus on critical data regions, enhancing model interpretability and diagnostic accuracy [68]. This combination demonstrates the versatility of ML models in comprehensively addressing the challenges in MCI to AD conversion classification.

From Table 2, it was found that deep learning approaches possess the ability to automatically extract intricate patterns, particularly when dealing with brain-related information. Their strength lies in excelling at learning hierarchical features directly from raw input data. However, a notable drawback is their dependency on a substantial volume of datasets for optimal performance. Additionally, these methods necessitate extensive training time and resources, which may pose challenges in terms of efficiency and scalability. Another issue commonly associated with them is the tendency to experience overfitting, limiting their capacity to generalize well to new or unseen data. A notable advantage of transfer learning methods is their ability to make use of knowledge acquired from a single task to enhance performance on another, proving particularly effective when labeled data for the target task is limited. However, a key limitation lies in their domain-specific nature, restricting their applicability to certain contexts. Additionally, interpretability can pose a challenge, making it difficult to comprehend the underlying reasoning of the model. Despite these drawbacks, the capability to transfer knowledge between tasks and excel in scenarios with limited labeled data highlights their utility in various ML applications.

To increase the precision of early AD diagnosis, B. Cheng et al. [57] suggested a unique Multi-domain Transfer Learning (MDTL) architecture that makes use of data from both CSF and MRI. This model finds traits that are instructive from both MRI and CSF data, offering insights into the underlying physiological changes associated with AD progression. The framework of this model can be modified to incorporate other data modalities beyond MRI and CSF, providing flexibility for future enhancements. The group-level forecasts that were the subject of this work can be transformed into personalized models that take into consideration individual differences in risk factors and disease development. Additionally, additional validation with progressively larger and more diverse populations is necessary to verify the model’s general performance. Using a multimodal and multiscale deep neural network (DNN) framework, D. Lu et al. [58] presented a novel method for early AD diagnosis. The method integrates structural MRI and FDG-PET images to improve complementary information and offers a comprehensive view of disease progression across multiple scales. This model approach eliminates the need for manual feature engineering and potentially provides insights into the underlying disease mechanisms. When multiscale information is integrated, features at various anatomical scales can be captured, leading to more detailed representations of the course of disease than when single-scale techniques are used. It necessitates the creation of customized models that account for individual differences in risk variables and illness progression. It also needed clinical validation and confirmation that the model’s performance is generalizable across various demographics and imaging techniques. W. Lin et al. [60] used CNNs for analyzing MRI images to predict AD conversion particularly considering brain areas that cause AD and are primarily responsible for the predictions. Furthermore, Leave-One-Out cross-validation was applied, which increased the reliability and generalizability of the results. Because of the study’s reliance on a tiny dataset, questions are raised on how broadly the findings may be applied. Because of its “black box” character, it is difficult to comprehend how it gathers features and generates predictions that call for model interpretability. Thorough clinical validation in real-world scenarios is imperative before the model can be used in a clinical trial. A novel and promising method for diagnosing brain diseases was developed by M. Liu et al. [63] using the combination of landmark-based feature extraction and deep multi-instance learning. This method profited from the spatial information provided by landmarks as well as the robust learning capabilities of deep models. Large datasets can typically be handled by this deep learning framework in an efficient manner. Aside from this, the model’s decision-making process may be somewhat interpretable due to the utilization of landmarks. It is essential for medical applications. This article should provide justification for the selected landmarks and possibly investigate different or automated selection techniques. Overfitting is a common problem with deep learning models, particularly when there are limited data. To make sure the model performs effectively when applied to previously untested data, the study should provide strong validation techniques. R. Casanova et al. [64] analyzed MRI scans from over 1500 participants across three large imaging databases: WHIMS-MRI, ADNI, and CHARGE. A novel ML technique called Group Factor Analysis (GFA) was used to extract a shared anatomical signature of AD across the different datasets, despite variations in MRI acquisition and processing protocols. The “AD Spatial Risk Factor” (AD-SRF), captures patterns of brain tissue shrinkage associated with AD progression and it is also possible to forecast the conversion of AD in people with MCI. However, it needs to develop individual-level prediction models that account for patient heterogeneity and diverse risk factors. It needs an in-depth understanding of the underlying biological mechanisms. The study focused only on MRI data; it would give better results by exploring the integration of other imaging modalities like PET or CSF analysis. S. Basaia et al. [61] utilized deep neural networks (DNNs) to automatically classify MCI and AD directly from single MRI scans which offers a potentially faster and less invasive alternative to traditional diagnostic methods. Analyzing data from just one MRI scan simplifies the diagnostic process and reduces costs. It maps to visualize which brain regions, offered valuable insights into the neuroanatomical basis of AD and MCI. This study focused on group-level classification, it needs personalized prediction models that account for individual variations in disease progression and risk factors. It also required clinical validation in real-world settings before integrating into routine diagnostic procedures. G. Lee et al. [59] combined deep learning models on MRI, PET, and CSF data, to diagnose a more comprehensive way of AD progression than single-modality approaches. Longitudinal prediction of AD progression offered valuable information for patient monitoring and treatment planning. This study used attention mechanisms to identify brain regions and biomarkers most relevant to AD predictions. However, combining data from different modalities and scanners can introduce challenges; it requires standardizing data acquisition and preprocessing techniques for robust model performance. F. Gao et al. [62] used AD-NET that incorporates an age adjustment dynamic feature, allowing the model to better account for the natural progression of brain aging and improve prediction accuracy for MCI to AD conversion. AD-NET achieves significantly higher scores compared to traditional ML models and other deep learning approaches, making it more effective in identifying MCI patients who are at risk of becoming AD patients. This proposed mechanism offers some insight, but deep learning models remain complex black boxes that require further explainable AI techniques to understand the model’s predictions and age-adjusting mechanisms. It required standardizing data acquisition and preprocessing techniques that are crucial for robust model performance and generalizability. Y. Wei et al. [65] used a self-supervised technique due to a lack of diffusion MRI data for standard anatomical MRI scans. A contrastive learning approach is used to create brain networks from training deep learning models. They showed a unique approach by focusing on modeling healthy aging trajectories in brain networks to understand how normal aging deviations can offer valuable insights into early disease stages. It primarily focused on MRI data; it acknowledged the potential for other modalities like PET or CSF for a more comprehensive understanding of AD progression. However, it is required to validate with other datasets which is critical for ensuring generalizability. It is needed to fully interpret the model’s decision-making process and identify specific brain regions contributing to the predictions for analyzing deviations from healthy ageing trajectories. It primarily focuses on single-time point predictions; still, utilization of longitudinal data could provide higher insights into disease progression and improve prediction accuracy. S. El-Sappagh et al. [66] combined transfer learning and contrastive learning in a two-stage model with improved pre-trained features from medical images and then refined them for specific AD detection, potentially improving accuracy and efficiency. They basically focused on cropped patches from the hippocampus for more targeted feature extraction. 3D Grad CAM (Gradient-based Class Activation Mapping) was also utilized for insights into the brain regions contributing to the model’s predictions and was considered a unique approach in estimating the time frame for MCI conversion to AD. Further investigation on 3D Grad-CAM to fully understand how the model differentiates between AD, MCI, and healthy individuals, especially regarding the prediction of MCI conversion time is needed. This study focused on group-level analysis; however, it is required to develop personalized models that account for individual variations in disease progression, risk factors, and conversion time. It also required a larger dataset for MCI conversion time prediction since conversion time might be limited by the relatively smaller dataset and for clinical validation. P. Lu et al. [67] used an approach of combining feature selection and domain transfer learning in a two-stage model for efficient feature extraction from small MCI datasets. Targeting both pMCI and sMCI provides a better understanding of disease progression and potentially improves prediction effectiveness. The model of this paper could be used for personalized prediction by incorporating individual-level features, which could significantly improve clinical utility. This model performed well for pMCI, but not for sMCI, so it is required further study to improve prediction accuracy for this sub-group. Additional interpretability efforts are required to understand how the model differentiates between pMCI, sMCI, and healthy individuals. It also required validation with other datasets to ensure generalizability and robustness. F. Ren et al. [68] employed Deep Zero-Shot Transfer Learning (DZTL) which is unsupervised learning to accurately forecast the straight progression of MCI to AD from MRI scans. This study utilized ResNet architecture with Deep Subdomain Adaptation networks (DSANs), offering some degree of interpretability by visualizing the brain regions, contributing to the model’s prediction. DSANs provide some insights that require further research to fully understand the model’s decision-making process and identify the specific features for its predictions. This study focused on group-level prediction; there are needs to develop personal models that account for individual variations in disease progression, risk factors, and conversion patterns. While the study uses single time point MRI scans, incorporating longitudinal data could potentially improve prediction accuracy and provide insights into disease progression over time.

It was observed from the study of deep learning and transfer learning for classifying AD and MCI that CNN is the most commonly used model and ADNI is the most common dataset. An analysis of accuracy obtained from different models in the literature on ADNI MRI data is depicted in Figure 3.

It was observed from the study of deep learning to classify AD from MCI that there are some limitations including dependency on large datasets, overfitting, and interpretability. Deep learning models typically require vast amounts of data to achieve optimal performance. In the context of AD diagnosis, acquiring such large datasets with detailed annotations can be challenging and time-consuming. This limitation restricts the applicability of these models in settings with limited data availability. Deep learning models, with their complex architectures and numerous parameters, are susceptible to overfitting, especially when trained on small or imbalanced datasets. This occurs when the model learns the training data too well, leading to poor generalization of unseen data. Techniques like regularization, data augmentation, and early stopping can mitigate this issue, but it remains a significant challenge. Deep learning models are often considered black-box, making it difficult to understand how they arrive at their predictions. This lack of interpretability hinders trust and clinical adoption, as it is essential to comprehend the underlying decision-making process for medical applications.

While the aforementioned techniques demonstrate promising results in AD diagnosis, their successful translation into clinical practice requires careful consideration of several factors:The performance of these models needs to be rigorously evaluated in real-world clinical settings to assess their generalizability and clinical utility.The cost–benefit analysis of implementing these techniques is crucial. While they may improve diagnostic accuracy, the associated costs for hardware, software, and expertise should be weighed against the potential benefits.The use of patient data raises ethical concerns related to privacy, data security, and informed consent. Robust data protection measures must be in place.As mentioned, the black-box nature of some models can hinder clinical adoption. Efforts to improve model interpretability are essential for building trust among clinicians.

### Accomplished the Research Objectives with Success


**What are the geographical origins (country) of Dataset of the individuals involved in the research studies?**


Among the selected studies, it is notable that the ADNI dataset is frequently employed [69]. The ADNI dataset encompasses participants from various regions within the United States and Canada. Specifically, studies utilizing the ADNI dataset are referenced within the range of [37,38,39,40,41,42,43,44,45,46,47,48,49,50,51,52,53,54,55,56,57,58,59,60,61,62,63,64,65,67], and these investigations exclusively involve participants from North America.

Additionally, a limited number of studies are observed to have conducted research with participants from Europe. Specifically, four studies were conducted on participants from Europe [38,40,62].

Furthermore, there is a single study that comprises study participants from Portugal [40] and another that includes participants from Germany [38].

Notably, an intercontinental study is mentioned in reference [61], where participants from Milan, Italy, are incorporated. It is worth noting that this Milan-based study is combined with the ADNI dataset.

These findings illustrate the diverse geographical sources of study participants, with a predominant focus on North America, particularly through the use of the ADNI dataset, as well as limited representation from Europe and other locations.

2.
**Which types of data modalities are employed as features?**


This question investigates the data modalities used as features for prediction purposes, categorizing them into single-modality and multiple-modality approaches.


**Single-modality:**


In several studies, a single modality of data is employed for prediction. For example, Magnetic Resonance Imaging (MRI) is utilized, focusing on features like hippocampal volume, entorhinal volume, amygdale distance, intracranial volume, cortical thickness, distinguishing voxels, gyrus height, network-based features, whole MRI image patches, and 2D slice intensity features [39,42,43,44,45,53,56,59,60,62].

Similarly, PET data alone are used for prediction, involving features like values of metabolic intensity derived from unprocessed PET scans [41]. PET with CNN can also be used to early detect AD [70]. There are a lot of studies going on florbetpair which is used as PET features for distinguishing AD from normal control. This imaging technique is adept at discerning high cognitive memory in individuals with robust brain metabolism by utilizing radioactive tracers such as fluorodeoxy glucose (FDG) [41,69]. EEG data are instrumental in studies that aim to construct a graph structure representing functional connectivity within EEG signals [49].


**Multi-modalities:**


In this paper, researchers also employ multiple modalities of data for prediction, with 12 studies showing this approach. For instance, in one investigation, a mix of unstructured data from MRI, PET, and clinical data is used, incorporating features such as temporal gyrus and hippocampus volumes from MRI, independent component analysis (ICA) from PET, and neuro-psychological assessment information [42]. Another study combines MRI and PET data [58], while another integrates functional MRI (fMRI) and MRI data [46]. The latter involves extracting features from MRI using the FREESURFER program and creating a connection matrix for 93 areas of interest (ROI) [46].

Several studies employ an amalgamation of structured information from several modalities, such as MRI and CSF data [38], or MRI and demographic or psychological test data [40].

In one study, a model of the aging health trajectory is crafted using an organized dataset with several dimensions [65].

In five researches, a combination of information that is both structured and unorganized from different modalities is utilized. MRI, PET, neurological, and clinical data are combined, with MRI data including voxel-based and volume-based morphometry, neuro-psychological data (e.g., MMSE), and clinical data like demographics and medical history [42].

Within an additional investigation, a mix of MRI, PET, CSF, demographic, and neuropsychological information was utilized. Texture values captured using neural networks, reflecting gray and white matter intensity, are combined with genetic data (e.g., APOE4) and MMSE [37,52,55].

Lastly, in a study focused on sMCI and pMCI converters, whole brain image patches and cognitive assessment data are employed [64].

These findings highlight the diverse use of both single and multiple data modalities for prediction in MCI to AD conversion studies, enabling a comprehensive analysis of various features.

3.
**How many of the studies incorporate data obtained through longitudinal follow-up, and what is the time frame of the data follow-up considered in literature?**


This question investigates the use of follow-up data from longitudinal studies and the associated duration of this data, as well as how many follow-up appointments.


**6-Months Follow-Up:**


Several studies focus on a 6-month follow-up period, including references [37,39,41,42,43,44,45,46,51,52,53,54,55,57,58,60,63]. Notably, one study encompasses complete 2-year data [41]. Additionally, researchers across multiple references make use of 6-month data within the context of complete 3-year follow-up [37,39,42,44,46,51,52,53,57,58,59,60,61,63], and one study employs complete 3.5-year data [43]. Complete 4-year follow-up data feature in specific references [45,55].


**3 Years, 12 Months:**


In one study referenced as [38], the 12-month follow-up data duration is followed for three years (totaling three follow-up visits).


**1 Year, 5 Months:**


A particular study, denoted as [40], incorporates 1-year follow-up data and extends this observation for an additional 5 years, resulting in a total of five follow-up visits.


**Unclear Follow-Up Data:**


In studies cited as [47,48,49,56,62,64,65], the details regarding follow-up data are not explicitly specified.

4.
**What spectrum of transitions from MCI to AD is recognized and documented in the reviewed papers?**


This question investigates the MCI to AD Conversion


**Six Months:**


The shortest duration of conversion identified while reviewing the paper is 6 months [59].


**A Year:**


Four investigations predict MCI to AD conversion within a calendar year duration [45,48,58].


**Three Years:**


The majority of studies concentrate on predicting MCI to AD conversion within a 3-year period [37,39,41,42,46,51,52,53,55,57,61,63].


**Five Years:**


The longest time frame for predicting MCI to AD conversion, spanning five years, is examined in a specific article [40].


**Absent range of conversion:**


In some studies, such as [43,60], there is no specific time frame specified for the MCI to AD conversion. These investigations focus on predicting the conversion itself without pinpointing when it occurs.


**Unclear Conversion Range:**


The MCI to AD transformation span is not explicitly defined in the research article referenced as [47,48,49,54,56,59,61,62,64,65]. These findings provide insights into the diversity of conversion timeframes used in the examined studies and their relevance for healthcare decision-making.

5.
**What is the number of studies that explored the utilization of traditional ML approaches to predict the early transition from MCI to AD?**


This research question addresses the utilization of Traditional ML techniques in the papers.


**Traditional ML Algorithms:**


Several ML algorithms have been employed by researchers in the studies. SVM is notably used as a final classifier for the prediction of sMCI and pMCI detection. Initially, deep neural networks, including CNN or variants of CNN, are applied to extract important aspects from unstructured information. Subsequently, these retrieved characteristics or features are input into the SVM classification. SVM is implemented in the following papers: [37,38,39,42,45,53,55,58].

The Least Absolute Shrinkage and Selection Operator (LASSO) are utilized for the selection of features as well as categorization purposes in specific papers [41]. A feature selection framework is developed to select pertinent features, and the classifier is put into practice for the classification of sMCI and pMCI patients [41].

Models of surviving event time detection based on Cox are employed in studies referenced [51,52]. For the purpose of classifying and forecasting sMCI and pMCI patients, a Cox regression model is utilized; leveraging unstructured data gathered for the research using CNN models [51]. Likewise, a Cox hazard model is used to categorize patients with sMCI and pMCI helping identify a patient’s category based on the Cox probability value [52].

Longitudinal classifiers are used to detect sMCI and pMCI patients in two studies [40,43]. In one study, researchers propose a Mixed Effects Model for forecasting patients with pMCI and sMCI, taking into account patients’ visits at different intervals. Additionally, a method based on sliding windows is employed for classifying sMCI and pMCI patients, which assesses the impact of a single visit over another estimating a patient’s category [43].

These findings demonstrate the distinct application of ML and DL techniques in the examined papers, highlighting the various methodologies used in the context of MCI to AD conversion prediction.

6.
**In literature, how many deep learning and transfer learning models were employed to help for early MCI to AD conversion prediction?**


This research question addresses the utilization of deep learning and transfer learning algorithms.


**Deep Learning Algorithm:**


Architectures of deep learning neural networks are harnessed as classifiers and feature extractors in studies cited in references [46,47,48,49,54,59,60,61,62,64,65]. CNN architectures are extensively used to extract aspects of neuroimaging intensity, such as White Matter and Gray Matter intensity, as seen in references [46,47,48,49,54,61,62,65].

A combination of CNN and Recurrent Neural Network (RNN) is employed by researchers in studies [59,60,61]. These models are proficient in extracting spatial intensity-based features using deep CNN architectures and subsequently utilizing deep RNN structures to provide temporal data from the retrieved unstructured data attributes [59,60,61].

7.
**Under what specific conditions or circumstances do the ML models tend to perform better in predicting MCI to AD conversions?**


The range of performance indicators such as specificity, sensitivity, and accuracy is analyzed, and insights into which types of features and models perform better results are provided:


**Ideal Specificity, Sensitivity, and Accuracy:**



**Overall Performance:**


Across all studies, the reported ranges of accuracy, sensitivity, and specificity are as follows: 65–96%, 42–95%, and 53–100%, respectively.

For single-modality data, the ranges are accuracy 65–92%, sensitivity 42–95%, and specificity 53–95.15%.

For multi-modality data, the ranges are accuracy 70–96%, sensitivity 51–94%, and specificity 70–100%. These results indicate that models generally perform better when there are several data modalities.


**Measure Performance in Relation to Modalities:**


**MRI data alone:** The range of accuracy, sensitivity, and specificity is 65–83%, 69–95%, and 53–90%, respectively.

Free Surfer intensity values (used in some studies) have shown better performance.

PET data alone achieved an accuracy of 83%, sensitivity of 87%, and specificity of 78%.

Unstructured raw MRI data extraction through deep learning CNN reported results below 85%.

SVM on unstructured data exhibited a better specificity in the range of 77–96%.

Structural Volume Ratio and Geodesic Length (used in one study) reported the highest results, all below 90%, which is an encouraging outcome.

A study that used neuropsychological data alone on Portuguese participants attained an accuracy of 76%, sensitivity of 56% and specificity of 76%.


**Benchmarking Performance across Populations:**


Mixing ADNI and MILAN study participants using MRI data alone did not show better performance, with accuracy, sensitivity, and specificity of 74%, 75%, and 75%, respectively. This suggests the need for further research in developing algorithms for more generalized populations.

No studies have used multi-modal data for early prediction of MCI to AD conversion in intercontinental study participant populations.

Two studies conducted on German and Portuguese participants used different modalities for prediction tasks, with the study on German participants (MRI data and CSF) performing better, with accuracy, sensitivity, and specificity of 82%, 85%, and 90%, respectively.


**Performance in Relation to the Follow-Up Timeframe:**


Following eight follow-up data points for 6 months achieved a high accuracy of 92%, which is slightly better than following 6-month follow-up data for 3 years. In general, better results are reported when a shorter duration of follow-up periods (6 months) is used.

The predicted performance of MCI converters in relation to their range:

For 6-month conversions, one study reported an accuracy of 74%, sensitivity of 81%, and specificity of 71%. Better results, with accuracy, sensitivity, and specificity of 85%, were reported for 3-year MCI to AD converters in four studies.

This analysis provides insights into the performance of different features, models, populations, follow-up periods, and MCI conversion ranges. It underscores the potential of multi-modal data and the need for further research in diverse populations and longer follow-up periods.

## 4. Critical Evaluation of Different Datasets Used in Literature

The ADNI study emphasizes high-quality data collection and standardization across multiple sites through the evaluation and selection of 3D T1-weighted sequences, post-acquisition corrections, and continuous monitoring of scanner performance. However, challenges such as inconsistent readout gradients, noisy images with certain headcoils, and limitations in older imaging systems affected data quality. While the ADNI dataset includes a broad spectrum of participants with varying cognitive statuses, it lacks sufficient demographic diversity, with limited representation mainly from North America of minority groups and younger age cohorts. This lack of diversity and a bias towards higher socioeconomic status and education levels among participants limit the dataset’s generalizability. Despite these limitations, the standardized MRI protocol and data processing pipeline developed by ADNI have effectively minimized technical variability, serving as a valuable model for multi-site studies. Addressing the dataset’s demographic and representativeness gaps is essential for enhancing the applicability of research findings in Alzheimer’s disease and related disorders [15].

The AIBL study, which recruited over 1000 individuals aged 60 and above to study Alzheimer’s disease (AD), provides a valuable resource for advancing research in this field. However, it is important to critically evaluate the limitations and potential biases of the dataset. There is a potential for self-selection bias in recruitment, as participants who volunteer for such studies may have different characteristics compared to the general population. The “healthy controls” group may not be truly representative of a healthy population, as they had a range of risk factors for AD. The study’s cohort is predominantly Caucasian, limiting the generalizability of the findings to other ethnic groups. Furthermore, there is a need to consider the geographic and socioeconomic diversity of the participants, which may influence the study outcomes. Despite these limitations, the AIBL study boasts high data quality through methodologies, including comprehensive cognitive, biomarker, and imaging assessments. The study’s diverse range of assessments and longitudinal follow-up enhance its robustness, yet careful consideration of the demographic representativeness and potential biases is crucial for accurately interpreting the findings and their applicability to broader populations. Future expansions should aim to include more diverse participants and explore additional biomarkers to address these limitations and enhance the dataset’s utility in AD research [16].

The OASIS-3 dataset, a valuable resource for studying healthy aging and dementia, offers extensive longitudinal data on a well-characterized cohort. However, researchers should consider its limitations and potential biases, including sampling bias from recruitment at the Washington University Knight Alzheimer Disease Research Center, a focus on participants with a family history of Alzheimer’s, and a lack of standardized visit structure. Additional concerns include heterogeneity in imaging data due to scanner differences, demographic underrepresentation with a predominantly Caucasian cohort, age range limitations, and variability in cognitive and clinical assessment tools, and exclusion criteria that may not fully represent the older population with multiple comorbidities. Despite these issues, the dataset benefits from data processing, quality control, diverse participant cohorts, and multi-modal data types, although its geographic representation and focus on genetic predispositions might limit broader applicability. Future expansions with more diverse participants and additional biomarker data could address these limitations and enhance its utility for Alzheimer’s research [17].

The MIRIAD dataset, which provides longitudinal MRI scans of Alzheimer’s disease patients and healthy controls, is a valuable resource for Alzheimer’s research, especially in developing and validating techniques for measuring brain atrophy. However, it is important to critically assess its limitations and potential biases. One potential limitation is the sample size, with only 46 Alzheimer’s patients and 23 controls, which may limit the generalizability of findings. Additionally, the dataset may be subject to selection bias, as the control group primarily consists of the patients’ spouses or careers, potentially introducing a homogeneity that does not reflect the general population. The dataset’s focus on a narrow age range and specific diagnostic criteria further restricts its representativeness. Moreover, the study’s reliance on a single MRI scanner and radiographer enhances consistency but may limit the applicability of findings across different imaging setups. The lack of explicit discussion on ethnic and socioeconomic diversity also raises questions about the dataset’s overall representativeness. Despite these limitations, the MIRIAD dataset benefits from high-quality controlled imaging data and a well-defined protocol, which enhances its reliability for intra-subject comparison over time. Researchers should consider these factors when interpreting findings and be cautious about generalizing results to broader populations. Future studies could improve representativeness by including more diverse participants and utilizing multiple imaging sites to enhance the dataset’s robustness and applicability [18].

J-ADNI dataset’s key issues include the reliance on automated segmentation techniques without manual correction, which may introduce inaccuracies in brain structure measurements, and the absence of neuropathological confirmation for diagnoses, potentially compromising data accuracy. The demographic focus on Japanese adults aged 60–84 limits generalizability to other ethnic groups, while the exclusion of individuals with significant comorbidities or those outside the specified age range may introduce selection bias. Additionally, the study’s failure to account for attrition rates in sample size estimations and the reliance on specific cognitive tests that might not capture all dimensions of cognitive decline could affect the study’s validity. Future research should focus on expanding demographic representation, incorporating neuropathological data for long-term follow-up, and developing strategies to minimize dropout and systematically analyze its impact, thereby strengthening the reliability of findings and guiding clinical trial design. Addressing these aspects will provide a more balanced and comprehensive view of the dataset’s strengths and weaknesses [19].

The Japanese Alzheimer’s Disease Neuroimaging Initiative (J-ADNI) dataset is a valuable resource for longitudinal studies on Alzheimer’s disease, providing extensive neuroimaging, biomarker, and clinical data. However, it is essential to critically evaluate its limitations and potential biases. A key limitation is the potential for selection bias, as the participants might not represent the broader population of individuals with Alzheimer’s disease due to recruitment methods and specific inclusion criteria. Additionally, the dataset’s demographic characteristics, including age, gender, and socioeconomic status, are not thoroughly detailed, which raises questions about its overall representativeness. The use of standardized protocols compatible with the US-ADNI study enhances the dataset’s utility for comparative research, yet there are differences in population genetics and the applicability of findings across different populations. Furthermore, the paper does not discuss potential limitations related to the exclusion criteria, such as participants with significant comorbidities, which could affect the generalizability of the results. The lack of explicit mention of the diversity in the study sample, including ethnic and socioeconomic backgrounds, is another concern that may limit the generalizability of the environmental factors, and healthcare systems between Japan and the US may influence the findings. Despite these limitations, the J-ADNI dataset’s methodological approach, including the use of advanced imaging techniques and biomarker assessments, provides high-quality data for Alzheimer’s research. Future studies should aim to address these limitations by ensuring a more diverse and representative sample and by explicitly discussing potential biases and their implications for the generalizability of the findings [20].

## 5. Research Challenges

Identifying the correct and precise biomarkers from neuroimaging data poses a challenging task for researchers, especially when targeting specific populations or regions. The complexity increases when researchers aim to pinpoint the most accurate biomarkers crucial for distinguishing between MCI to AD converters and non-converters. Additionally, unfolding relevant features responsible for classifying individuals in top MCI to AD converters and sMCI to AD non-converters from a large group of multimodal features poses a significant challenge. The task of clearly identifying MCI to AD conversion, particularly within a short timeframe such as 6 months to 1 year, is inherently difficult. Urgently within the research community to quickly identify rapid MCI to AD converters underscores the need for the development of robust longitudinal models to address this challenging task.

## 6. Future Direction

It will be interesting for additional experiments involving diverse populations from various countries, regions, or states. This is necessary for a broader collaboration among hospitals and doctors worldwide. There is considerable potential for research aimed at creating a model applicable to a generalized population. Given the limited studies and the lack of data addressing populations outside the USA, Canada, and Japan, there is a need for further research. More investigations have to concentrate on AD sufferers in many different regions of the globe for the comprehensive research study. Analyzing the parameters responsible for AD and understanding regional variations presents an interesting opportunity for research. Exploring significant parameters contributing to MCI to AD conversions across regions and countries can provide valuable insights. A critical challenge in early MCI to AD prediction lies in identifying parameters that yield effective contributions to the process of ML, emphasizing the necessity of robust choice of different feature techniques. Developing algorithms capable of discerning the optimal collection of attributes or features and allocating weights to them could enhance specificity. Additionally, predicting MCI to AD conversions during a designated window of timeframe, such as identifying patients who will quickly transition to AD, remains a challenging task, it is also necessary for further comprehensive studies in this direction. Additionally, research on wearable therapeutic devices such as EEG or fNIRS-based music therapy shall be conducted for early detection of MCI to AD transition [71].

Despite advancements in distinguishing between progressive pMCI and sMCI patients, challenges remain due to overlapping characteristics and the need for reliable biomarkers. Future research should focus on identifying novel biomarkers through multimodal approaches, integrating genomic, MRI, PET, and metabolomics data to improve differentiation between pMCI and sMCI. Standardizing data collection and processing protocols across studies can enhance model generalizability, while integrating explainable AI (XAI) techniques can improve model interpretability and clinical trust. Research should address questions such as: What novel biomarkers can enhance the differentiation between pMCI and sMCI? How can we standardize data protocols to ensure consistency across studies?

Furthermore, the variability in longitudinal data and the need for scalable, real-time machine-learning solutions present significant challenges. Future research should focus on developing robust methods for handling missing data and optimizing algorithms for real-time applications in clinical settings. Innovative longitudinal analysis methods, including continuous monitoring and adaptive learning models, can improve prediction accuracy. Additionally, fostering collaboration between researchers, clinicians, and regulatory bodies is essential for transitioning models into clinical trials. Key research questions include: How can we handle missing longitudinal data to ensure reliable predictions? What strategies can optimize machine learning algorithms for real-time clinical use?

## 7. Limitation of the Study

Certain limitations of the existing study are given as follows:(a)Limited Time Frame in Literature Review: The study confines its literature review to studies published from 2016 onwards. Future work should consider extending the comprehensive literature review on Alzheimer’s detection to include studies from 2013 onwards, covering at least 10 years. The expansion of the review is essential for providing a deep understanding of the evolution of models and how feature extraction methods used over time which could benefit new researchers in the field.(b)Focus on Specific Patient Groups: The study predominately concentrates on ML papers identifying sMCI and pMCI to AD, which is currently a challenging task. To broaden the scope and applicability, future research should extend the focus to include the identification of AD, MCI, and healthy control (HC) patients. This expansion would contribute to a more comprehensive understanding of ML applications at varying degrees of cognitive decline.(c)Limited Detail on Feature Extraction: The paper lacks a thorough description of the techniques used in feature extraction methods employed in this reviewed study. Further investigations should be necessary for comprehensive coverage of several feature extraction methods specifically to categorize sMCI and pMCI. A comprehensive understanding of feature extraction methods is essential for researchers and practitioners seeking insights into the technical aspects of model development.(d)In-Depth Exploration of Deep Learning and Transfer Learning Architectures: The study acknowledges a gap in providing a thorough breakdown of the different deep learning and Transfer Learning architectures for identifying sMCI and pMCI patients. Future researchers should focus on a thorough exploration and explanation of deep learning and Transfer Learning architectures employed in studies and why they are necessary, particularly those utilizing medical images. Understanding the role of deep neural networks in feature extraction is crucial for advancing the capabilities of detection models.

## 8. Conclusions

The identification of specific parameters that differentiate individuals with Mild Cognitive Impairment (MCI) who progress to Alzheimer’s disease (AD) remains unclear and undefined. This falls under the critical and effective feature selection algorithms capable of extracting optimal feature parameter sets from extensive datasets. Additionally, this review highlights a limitation in the existing literature, where there is an absence of experiments conducted across diverse hospital settings and populations in various countries. To address this gap, there is a call for extensive collaboration and data sharing on a large scale within the AD research community. Such collaboration would enable researchers to conduct experiments with variables drawn from diverse populations, ultimately facilitating the development of models that are more universally applicable, such as a single model that would benefit the generalized population within a specific country or continent. It is essential, in the present state of research, to create a more generalized model. Furthermore, a significant challenge in the current research is to identification of MCI to AD converters, particularly within a short timeframe (e.g., within 6 months), using a minimal number of follow-up data points. Successfully achieving this task requires innovative approaches and model designs that can obtain accurate predictions with limited datasets. Moreover, it is notable that employing multiple data modalities is yielding more promising outcomes. This emphasizes choosing distinctive characteristics from different data modalities to inform the design of models that can effectively integrate and interpret information from different sources.

## Figures and Tables

**Figure 1 diagnostics-14-01759-f001:**
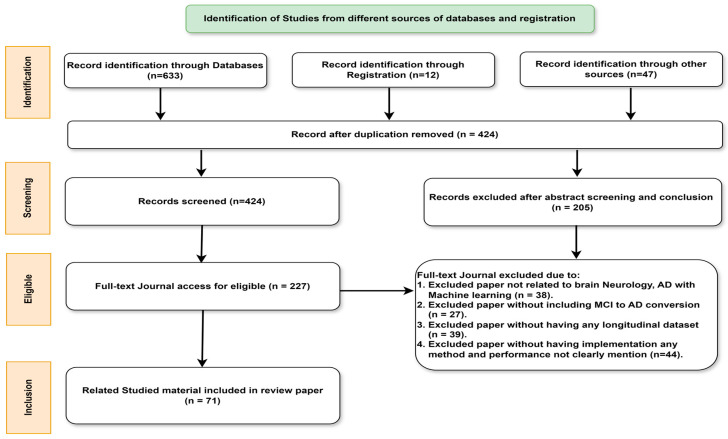
Journals Exclusion criteria used in the selection process.

**Figure 2 diagnostics-14-01759-f002:**
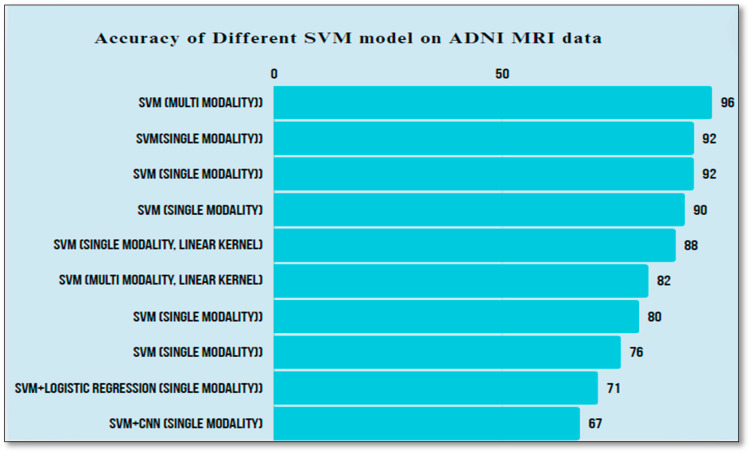
Accuracy analysis of different SVM models on ADNI MRI data.

**Figure 3 diagnostics-14-01759-f003:**
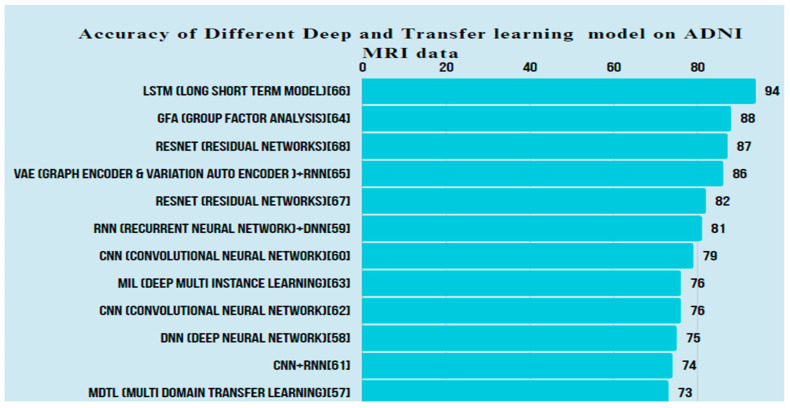
Accuracy analysis of different Deep and Transfer learning models on ADNI MRI data.

**Table 1 diagnostics-14-01759-t001:** Description of traditional ML models used in literature to classify AD from MCI.

Ref.	Dataset	Description of the Method	Number of Follow Up and Duration of Follow Up for MRI	MCI Range	ML	Results	Limitations
K. Ritter et al., 2015 [37]	ADNI	Single-modality and multi-modality Neuropsychological Measures (NM) and Structural Magnetic Resonance Imaging (MRI) morphometry, Manual Feature Ranking (MFR)	6 and 6 months	3 Years	SVM	Accuracy: 89.66% Sensitivity: 87.50% Specificity: 92.31% Precision: 93.33%	1. Conclusions might not be generalizable to a larger population. 2. Insufficient elaboration on challenges and biases introduced by the small dataset.
L. Frolich et al., 2017 [38]	ADNI	Multi-modality MRI hippocampal volume, CSFTau, A-Beta, no feature selection	3 and 1 year	2 Years and 1 Month	SVM with linear kernel	Accuracy: 82% Sensitivity: 85% Specificity: 70%	1. Findings might not be robust due to insufficient data. 2. Potential overfitting due to lack of appropriate cross-validation for smaller datasets. 3. Reliance solely on ROC curve might not provide a comprehensive evaluation.
X. Long et al., 2017 [39]	ADNI	MRI Single modality, Amygdala distance, no feature selection	6 and 6 months	3 years	SVM with linear kernel	Accuracy: 88% Sensitivity: 86% Specificity: 90%	1. Results may not be generalizable to a larger population. 2. Insufficient information about feature selection process. 3. Unclear discussion of potential confounding variables or methodological limitations.
T. Pereira et al., 2017 [40]	Lisbon, and the Neurology Department, University Hospital in Coimbra (Portuguese)	Multi-modality, Word recall test, Cancellation task verbal paired associate learning, Cube draw digit span, Raven progressive metrics, no feature selection	5 and 1 year	5 years	NB, RF, SVMRBF and SVM Poly,	Accuracy: 76% Sensitivity: 56% Specificity: 70%	1. Dataset from a specific Portuguese hospital might not represent the general population. 2. Lack of clarity on how the model makes predictions using time windows.
Y. Zhao et al., 2017 [41]	ADNI	PET single modality, metabolic intensity values, LASSO as feature selection	4 and 6 months	2 years	SVM	Accuracy: 83% Sensitivity: 87% Specificity: 78%	1. Limited generalizability of findings due to small dataset. 2. Risk of overfitting without proper cross-validation 3. Refinement needed in network construction to incorporate additional information.
K. Liu et al., 2017 [51]	ADNI	Multi-modality MRI (Temporal Gyrus, Hippocampus), PET (Both ICA) clinical variable, no feature selection	6 months	3 years	Cox Model	Accuracy: 84% Specificity: 86% Specificity: 82%	1. Impact of small dataset on model performance not fully addressed. 2. Combining multi-modal data using advanced techniques could enhance analysis.
Z. Sun et al., 2017 [42]	ADNI	MRI single modality, Structural volume ratio, Geodesic length, no feature selection	6 and 6 months	3 years	SVM	Accuracy: 92% Sensitivity: 95% Specificity: 90%	1. Reliance on ADNI dataset might limit applicability to diverse populations. 2. Insufficient detail about the specific relevance of anatomical development features.
G. Gavidia-Bovadilla et al., 2017 [43]	ADNI	MRI single modality, MRI cortical thickness, hippocampus volume, no feature selection	6 and 6 months	3 years	SVM	Accuracy: 76% Sensitivity: 70% Specificity: 81%	1. Lack of details about CSF observations at later stages, missing MRI data, and study population. 2. Limited explanation of model predictions and feature importance.
M. Gomez-Sancho et al., 2018 [44]	ADNI	MRI single modality, hippocampal volume, ICV, entorhinal volume, no feature selection	6 and 6 months	3 years	SVM and Regularized Logistic Regression	Accuracy: 71% Sensitivity: 53% Specificity: 53%	1. Tendency of SVM to overpopulate the pMCI class.
K. Kauppi et al., 2018 [52]	ADNI	Multi-modality MRI, atrophy score, MMSE, Genetic-PHS, no feature selection	6 and 6 months	3 years	Cox Proportional Models	Accuracy: 78% Sensitivity: 79% Specificity: 77%	1. Recruitment from memory clinics might introduce bias.
T. Shen et al., 2018 [45]	ADNI	MRI single modality, gray matter regions, automatic feature selection	8 and 6 months	1 Year above	SVM	Accuracy: 92% Sensitivity: 93% Specificity: 92%	1. Insufficient explanation of variables and their significance in CNN and SVM. 2. Lack of details about hyperparameter tuning in CNN.
S.H. Hojjati et al., 2018 [46]	ADNI	Multi-modality fMRI, fMRI-connectivity matrix for 93 ROI, Freesurfer features, MRMR and SFC for feature selection	6 and 6 months	unclear	SVM	Accuracy: 96% Sensitivity: 94% Specificity: 100%	1. Lack of justification for the choice of atlases for rs-fMRI and sMRI. 2. Limited discussion on reproducibility and generalizability of findings.
C.C. Luk et al., 2018 [55]	ADNI	Multi-modality MRI, genetic, neuro-psychological assessment, MRI-hippocampal volume, texture value of voxels, MMSE, APOE-4 MRI automatic feature selection	8 and 6 months	3 years above	Binary Logistic Regression	Accuracy: 93% Sensitivity: 86% Specificity: 83%	1. Insufficient visual explanations for texture changes. 2. Unclear discussion of potential biases and challenges in texture analysis.
W. Zheng et al., 2018 [53]	ADNI	MRI single modality, cortical thickness, surface area, volume surgical depth, gyrus height multi feature network, network multi feature network, automatic feature selection	6 and 6 months	3 years	Sparse Linear Regression (LASSO)	Accuracy: 65.61% Sensitivity: 70% Specificity: 58%	1. Need for replication on larger, independent samples. 2. Excluding hippocampal and subcortical regions might impact performance.
J.E. Arco et al., 2021 [47]	ADNI	Single modality, voxel MRI images	6 and 6 months	3 years	SVM	Accuracy: 80% Precision: 84% Sensitivity: 85% Specificity: 82%	1. Small sample size might limit applicability to larger populations. 2. Overlooking information about non-converters.
E.E. Bron et al., 2021 [48]	ADNI	Single modality, gray matter intensity	1 and 5 year	2–5 years	SVM + CNN	Accuracy: 67% Sensitivity: 68% Specificity: 66%	1. Insufficient details about external validation metrics and performance drop.
P.M. Rossini et al., 2022 [49]	NINCDSADRDA	Single modality PET and EEG, automatic feature extraction	unclear	unclear	SVM	Accuracy: 89% Sensitivity: 90% Specificity: 88%	1. Lack of details about dataset size and composition.
M. Inglese et al., 2022 [54]	ADNI	Single modality, whole MRI brain image	unclear	unclear	LASSO	Accuracy: 76.72% Sensitivity: 55.56% Specificity: 95.15%	1. Lack of extensive external validation on diverse datasets. 2. Inclusion of FTD and PD patients in control group might introduce bias. 3. Lower performance at 3T magnetic field strength. 4. High computational cost for pre-processing.
S. Liu et al., 2023 [50]	ADNI	Multi modality Neuroimaging data, CSF biomarkers, CFA, genetic biomarkers, and their combinations	Not clear	2 years	Unsupervised novel detection algorithms based on GMM, kNN, k-means	Adjusted F Score: kNN: 72.7%, GMM: 71.79%, ELM: 72.76%, SVM: 73.59%, RF: 47.71%. The area under curve: KNN: 85.51%, GMM: 84.53%, ELM: 84.73%, IF: 81.51%, SVM: 86.51%, RF: 78.23%	1. Need for a more generalized dataset. 2. Equal contribution of all modalities might not be accurate.
S. Park et al., 2023 [56]	ADNI	Single modality, segmented MRI brain image, automatic feature extraction	unclear	unclear	Logistic Regression	AUC 88%	1. Lack of information about original dataset size and characteristics. 2. Insufficient information about logistic regression parameters. 3. Reliance only on AUC for evaluation. 4. Lack of clarity on whether the dataset is longitudinal.

**Table 2 diagnostics-14-01759-t002:** Description of Deep Learning and Transfer Learning Algorithms used in literature to classify AD from MCI.

Ref.	Dataset	Description of the Method	Number of Follow Up and Duration of Follow Up for MRI	MCI Range	ML	Result	Limitations
B. Cheng et al., 2017 [57]	ADNI	MRI single modality,93 ROI GM	6 and 6 months	3 Years	Multi Domain Transfer Learning (MDTL)	Accuracy: 73% Specificity: 69% Specificity: 77%	1. The model does not account for individual variations in disease progression and risk factors. 2. The model needs to be tested on larger and more diverse populations to assess its generalizability.
D. Lu, et al., 2018 [58]	ADNI	Structural MRI Multi-modality FDG-PET, patch volume, mean intensity of GM 34 ROIs, automatic feature selection	6 and 6 months	1 year	Deep Neural Network (DNN)	Accuracy: 75% Sensitivity: 73% Specificity: 76%	1. The study focuses on group-level analysis without considering individual differences. 2. The model needs to be evaluated in a clinical setting to assess its practical utility.
W. Lin et al., 2018 [60]	ADNI	MRI single modality, intensity values automatic feature selection	6 and 6 months	3 years	CNN	Accuracy: 79% Sensitivity: 84% Specificity: 74%	1. The small sample size might limit the generalizability of the findings. 2. The black-box nature of CNNs hinders understanding of the model’s decision-making process. 3. The model needs to be tested in real-world clinical settings.
M. Liu et al., 2018 [63]	ADNI	MRI single modality, whole patches of image, automatic feature selection	6 and 6 months	3 years	Deep Multiple-instance Learning (MIL)	Accuracy: 76% Sensitivity: 42% Specificity: 82%	1. The justification for the selected landmarks is unclear. 2. Deep learning models are prone to overfitting, especially with limited data. 3. Robust validation techniques are essential to ensure model performance on unseen data.
R. Cas-anova et al., 2018 [64]	ADNI	Single modality, whole patches, MRI cognitive data	unclear	unclear	Group Factor Analysis (GFA)	Accuracy: 88% Sensitivity: 88% Specificity: 88%	1. The study focuses on group level analysis without considering individual differences. 2. Incorporating other imaging modalities could enhance the model’s performance. 3. Further research is needed to understand the underlying biological processes.
S. Basaia et al., 2019 [61]	ADNI	MRI gray matter, white matter intensity, automatic feature extraction	6 and 6 months	3 years	CNN + RNN	Accuracy: 74% Sensitivity: 75% Specificity: 75%	1. The study focuses on group level classification without considering individual variations. 2. The model needs to be evaluated in real world clinical settings.
G. Lee et al., 2019 [59]	ADNI	Multi-modality MRI, CSF-A Beta 42, Peptide, Tau Genetic-POE4 Neuropsychological MMSE, no feature selection	6 and 6 months	2 years	RNN + Deep Neural Network	Accuracy: 81% Sensitivity: 84% Specificity: 80%	1. Combining data from different modalities and scanners can be complex. 2. Deep learning models can be difficult to interpret.
F. Gao et al., 2020 [62]	ADNI	Single-modality, whole patches of MRI	unclear	unclear	CNN	Accuracy: 76% Sensitivity: 79% Specificity: 76%	1. The black box nature of deep learning models hinders understanding of the model’s decision making process. 2. Consistent data acquisition and preprocessing are crucial for reliable results.
Y. Wei et al., 2022 [65]	ADNI	Multi-modality brain MRI, cognitive data, Autoencoder to extract node feature + Cross model contrastive	4 and 6 months	1.5 years	Graph Encoder and variation auto encoder (VAE) + RNN	Accuracy: 86.1% Sensitivity: 88.5% Specificity: 83.33%	1. Incorporating other modalities could provide a more comprehensive understanding of AD. 2. Understanding the specific brain regions contributing to the model’s predictions is essential. 3. Using longitudinal data could enhance the model’s predictive power.
S. El-Sappagh et al., 2022 [66]	ADNI	Multi-modality MRI, cognitive scores, CSF biomarkers, neuropsychological battery makers and demographics	6 and 3 months	1.5 years	Long Short- Term Model (LSTM)	Accuracy: 93.87% Precision: 94.07% Recall: 94.07% F1-score: 94.07%	1. Personalized models are needed to account for individual differences. 2. A larger dataset is required for accurate MCI conversion time prediction. 3. Further research is needed to fully understand the model’s decision making process using 3D Grad-CAM.
P. Lu et al., 2022 [67]	ADNI	Single-modality, MCI dataset by adopting 3D CNN, automatic feature extraction	6 and 6 months	3 years	3D ResNet + MoCo	Accuracy: 82% Sensitivity: 79% Specificity: 85% AUC: 84%	1. The model needs improvement in predicting sMCI conversion. 2. Understanding how the model differentiates between pMCI, sMCI, and healthy individuals is crucial. 3. The model needs to be tested on other datasets to assess its generalizability.
F. Ren et al., 2023 [68]	ADNI	Single-modality MRI, 3D grey matter, elastic mix-up for augmentation	not clear	not clear	Unsupervised learning zero-shot learning 3D-Resnet + DsAN	Accuracy: 87.16% Sensitivity: 78.11% Specificity: 92.40%	1. The study focuses on group level prediction without considering individual differences. 2. Further research is needed to fully understand the model’s decision making process. 3. Incorporating longitudinal data could improve prediction accuracy.

## Data Availability

No new data were created or analyzed in this study. Data sharing is not applicable to this article.

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
