# Peer review of "Early Alzheimer’s Disease Detection: A Review of Machine Learning Techniques for Forecasting Transition from Mild Cognitive Impairment"

_diagnostics, 2024, doi:10.3390/diagnostics14161759_

Round 1

Reviewer 1 Report

Comments and Suggestions for Authors

This review examines machine learning approaches for predicting Alzheimer's disease conversion from mild cognitive impairment to Alzheimer’s disease dementia using MRI, PET, and other biomarkers. It discusses various techniques, data modalities, and feature extraction methods, providing a comprehensive overview of current research and highlighting future directions.

Although the manuscript is of interest and merit, some Recommendations for Improvement should be addressed before publishing.

1)      The first abbreviations in the text should be marked throughout, such as AD, PET, EEG, MEG, NIRS, TMS, etc  . Please pay attention to the writing standard and re-check the full text.

2)      In Tables 1 and 2, it would be helpful to add the year of each references in the first column and add another column to summarize the limitations of each cited work in this table.

3)      The third figure in the manuscript is not correctly numbered, it should be numbered as figure 3, as well as it is better to cite the reference that used the Deep learning model in the y-axis

Reviewer 2 Report

Comments and Suggestions for Authors

Your manuscript titled "Early Alzheimer's Disease Detection: An Organized Review of Machine Learning Techniques for Forecasting Transition from Mild Cognitive Impairment" provides a comprehensive overview of the current advancements in using machine learning (ML) for predicting the progression from mild cognitive impairment (MCI) to Alzheimer's disease (AD). The synthesis of various ML approaches, including deep learning and transfer learning, is thorough and well-organized. The inclusion of a wide array of data modalities and the discussion of different feature extraction methods enhance the depth of the review. However, some areas require further elaboration and clarity to improve the manuscript's overall impact.

Firstly, while the review covers a broad spectrum of ML techniques, it would benefit from a more detailed discussion on the limitations and challenges associated with each method. For instance, the dependency on large datasets for deep learning models and the potential for overfitting should be emphasized. Additionally, the manuscript could delve deeper into the practical implications of these techniques, such as their applicability in real-world clinical settings and the potential for integration into existing diagnostic workflows.

Secondly, the manuscript mentions various public datasets like ADNI, AIBL, and OASIS but lacks a critical evaluation of these datasets' limitations and potential biases. A discussion on the quality, diversity, and representativeness of these datasets would provide a more balanced view. Furthermore, including a comparative analysis of the performance of ML models on different datasets could highlight the generalizability and robustness of these techniques.

Thirdly, the review could be enhanced by a more explicit discussion on the future research directions and open challenges in this field. While some future directions are mentioned, a dedicated section summarizing key challenges and proposing specific research questions or methodologies to address these gaps would provide valuable guidance for future research.

Lastly, the readability of the manuscript can be improved by ensuring consistent terminology and clearer explanations of technical terms and concepts. Including more visual aids such as tables, charts, and diagrams to summarize key points and findings would also aid in better comprehension.
